# Filtration Process and Alternative Filter Media Material in Water Treatment

**Anna Cescon and Jia-Qian Jiang *** 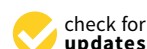

Department of Civil Engineering and Environmental Management, Glasgow Caledonian University, Scotland G4 0BA, UK; anna.cescon.11@gmail.com
* Correspondence: jiaqian.jiang@gcu.ac.uk

**Abstract:** In drinking water treatment, filtration plays an important role in the multi-barrier approach employed for the removal of pathogens. The presence of suspended solids and other particulate matter in water increases the resistance of most microbes to disinfection. Therefore, high performance in the removal of particles achieved by granular filtration can increase the disinfection efficiency. Although sand is one of the major filter media, alternative media have been developed and used in recent years. In this review, the performance of alternative media is compared with traditional sand/anthracite for drinking water treatment. Advantages in the use of alternative media, especially glass media, have been found, including high filtration performance in removing residual particles and turbidity, minor modification requirements to the existing filtration configuration and slow head loss development. However, before the employment of them in industry, additional tests are recommended. In particular, full scale tests with variations in the operating conditions and analyses of pathogen removal should be performed. Moreover, this paper reviews the filtration processes and operating configurations which provide overall references to those who are studying and working in the field of water technology and treatment. In this paper, legislations/standards of safe drinking water are summarized as they are the driving force of developing new treatment technologies; mathematical modules for predicting the media filtration performance are briefed. Finally, future work on the application of alternative filter media is recommended.

**Keywords:** alternative filter media; filtration process; recycled glass media; water treatment; waste water treatment

## 1. Introduction

Safe water supply is essential to maintaining human life and sustain ecosystems and community societies. Drinking water treatment is a complex issue, and the treatment chain comprises several processes, which vary depending upon the legislation, the removal targets of contaminants and the costs associated with them. In 1854, it was discovered that a cholera epidemic spread through water. The outbreak seemed less severe in areas where sand filters were installed. British scientist John Snow found that the direct cause of the outbreak was water pump contamination by sewage water. He applied chlorine to purify the water, and this paved the way for water disinfection. This discovery led to governments starting to install municipal water filters (sand filters and chlorination), and hence the first government regulation of public water. Since then, among water treatment chains, filtration has constituted the centre of drinking water treatment—together with disinfection—for well over a century [1]. Filtration represents a barrier for some of the most common issues encountered in the water supply industry; colour, turbidity and harmful microorganisms being the main ones among them. Moreover, filtration plays an important role in the multi-barrier approach employed for the removal of pathogens. The presence of suspended solids and any particulate matter increases the

resistance of most microbes to disinfection. Therefore, a low particle removal performance by granular filtration can decrease the disinfection efficiency [2].

Pathogens and organic compounds are some of the main foci for water treatment in Scotland; as a consequence of the land and the livestock farming practiced in the region. For this reason, there is considerable interest in efficiency improvements. Traditionally, filtration employs granular media, such as sand, in either rapid or slow filters, depending upon the flow rate applied. In the second half of the 20th century, dual media configurations, including a layer of anthracite on top and occasionally an additional third thick layer of gravel, started to be employed. While these solutions are still widely used all over the world and have proved to be reliable and effective, recent changes to the legislation and a general push towards higher efficiencies are leading to research into other possibilities. These include not only modifications to the process and/or the application of new materials that allow higher treatment performance, but also lower costs and especially more energy efficient procedures. Concerns about climate change are pushing utilities to adopt energy savings policies and to reduce the $CO_2$ emissions to the minimum compatible with high quality production. Climate change also has an influence on the process itself, as it causes a variation in the concentration and in the physico-chemical characteristics of the contaminants. For example, increasing temperatures leads to increases in the rates of solubilization and the decomposition of the compounds, with changes to the features that make the removal of contaminants more difficult [3]. More control over the influent and the process characteristics could present a solution. An optimization of the processes could, however, provide optimal removal and compliance with the required effluent concentration, regardless of the load imposed by the influent.

For these reasons, researchers have been working towards improving the efficiency of the filtration processes which are currently employed. Several approaches have been taken. On the one hand, the filtration has been focused on the employment of more effective alternative media. On the other, modifications have been performed on the process itself, departing from the traditional granular media filtration towards the application of membranes and fibres. The latter solution, though becoming more widespread with a consequent reduction in costs, remains relatively expensive and is generally used in limited sites. Application is generally sought where specific issues related to the raw water make it desirable and where other, less costly, solutions have proved to be ineffective. Even in that case, however, rapid granular filtration is often performed as a roughing treatment to ensure the removal of contaminants that could endanger the integrity of the membranes.

This paper reviews the filtration process and its role in water treatment first. Secondly, in the Supplementary Materials, we provide reviews on the standards for safe drinking water, since water treatment targets to meet legislations of safe drinking water, which are a driving force of developing new treatment technologies. Moreover, we brief mathematical modules for predicting the performance of media filtration as they are one of the most important aspects of discussing filtration for water treatment. Finally, we focus on the cases of applying alternative types of filter media; which are more likely to be used in existing filtration structures without requiring major infrastructure changes.

## 2. Filtration for Water Treatment

### 2.1. Filtration Process

Filtration is one of the core processes in water treatment. The term refers to the removal, mainly by physical action, of suspended solids as the suspension flows through a bed packed by granular media. If a coagulant is added, colloids can be removed at the same time and the range of detained particles increases considerably. Filtration focuses mainly on turbidity, colour, microorganisms and particulates, whether already present in the water or formed via pre-treatment [4,5]. The particles involved are considerably smaller than the grain size, as shown on the right-hand side of Figure 1.

# Particle size (μm)

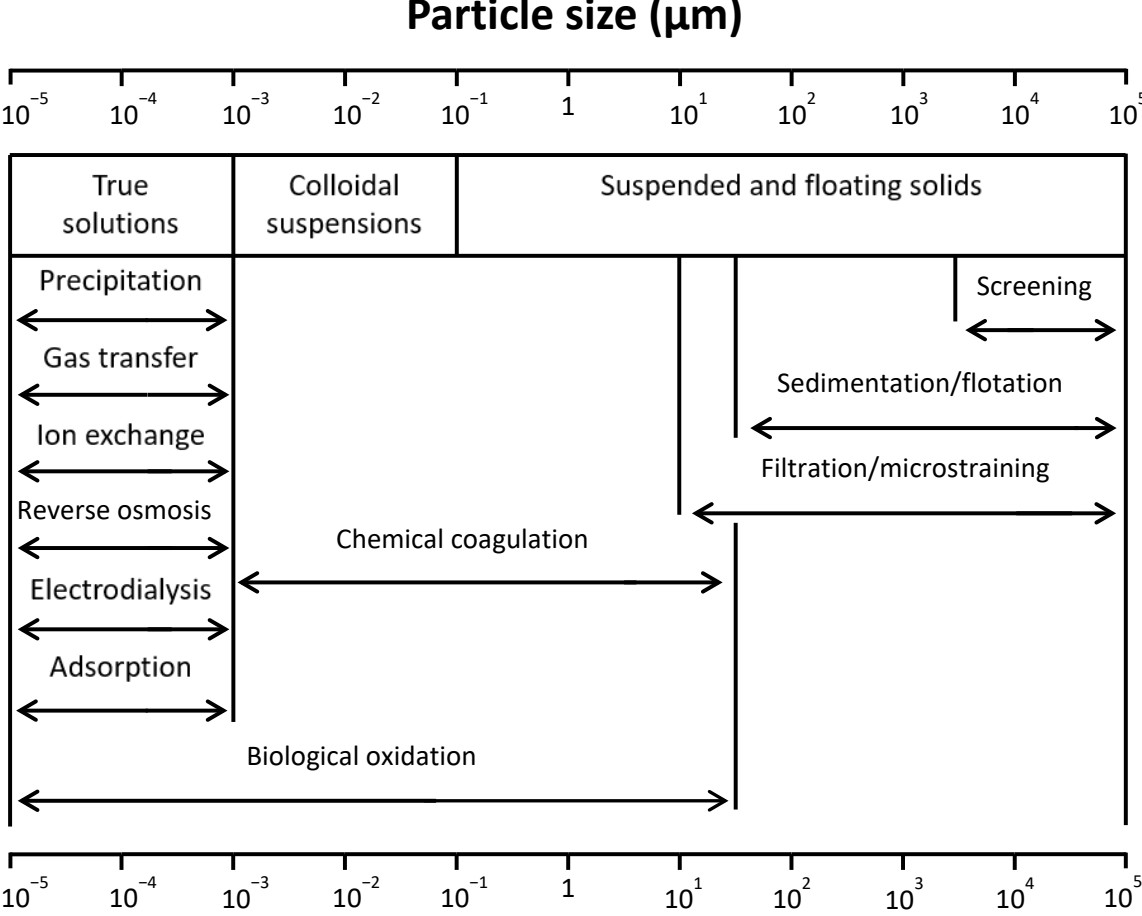

**Figure 1.** Range of effectiveness of the treatment processes (reprepared using the data from [6]).

Filter efficiency is determined by physical characteristics, such as grain size, shape, porosity, and bed depth/media grain size ratio. McGivney and Kawamura [7] suggest using L/de ratio for the design of filter beds where L is the depth of the filter bed (mm) and de is the effective size of the filter medium. L/de ranges between 1000 and 2000 for different filter beds—1000 for ordinary mono-medium sand and dual media beds, 1250 for regular tri-media beds (coal, sand, and garnet), 1250 to 1500 for coarse deep mono-medium beds in which de is 1.2 to 1.4 mm, and 1500 to 2000 for very coarse mono-medium beds in which de is 1.5 to 2.0 mm. It is also emphasized that pilot studies are recommended in the selection of filter depths when the medium is larger than 1.5 mm. The same authors [7] also recommend using L/de ratios that are increased by 15% to achieve filtered water turbidity less than 0.1 Nephelometric Turbidity Unit (NTU).

Filtration might occur as depth or as cake filtration. In the former, particles are caught in the pore system of the medium via attachment, while the second one involves the creation of a "cake" on the surface of the medium, with most solids being removed on the top [4]. Granular media filters act through depth filtration, with the majority of the filter media involved [8].

The filtration process consists of a transport stage, which takes particles closer to the filter media and an attachment stage that depends upon particle–surface interactions. As attachment mechanisms might cause the deviation of particles towards the surface of the grain, the two stages are not completely distinct [9,10]. Some authors, however, tend to discard the influence of surface forces when compared to the effect of transport mechanisms [11]. Particles might also undergo aggregation; they can form clusters that are more easily transported and deposited on the medium. Detachment is considered to be additional stage as well, causing particles to re-join the flow [12,13].

The study of deep bed filtration has relied on the interpretation of the filter bed as a group of single collectors; the efficiency is calculated considering the bed to be composed of uniform spheres acting as collectors [14]. The removal at any given plane at a certain distance from the surface of the media will be a function of the number of collectors located within that distance. This transforms the problem into transport and deposition of particles onto individual grains (trajectory analysis or microscopic model) [15]. The trajectory analysis is, however, valid only for a clean filter; the deposition of particles will vary the characteristics of the filter bed and the flow pattern. Removed particles will act as additional collectors for the particles subsequently reaching the bed [16]; their effect has to be included when calculating the efficiency. Some authors argue that these additional collectors can be more effective than the filter grains themselves [17].

During the process, several stages can be recognised (Figure 2). According to Jegatheesan et al. [10], an initial stage might be identified when the filter is still clean, followed by a second transient stage. During the transient stage, the filter performance is at first improved (ripening), then maintained throughout a subsequent working stage, and it finally deteriorates during breakthrough. The improvement in the performance is due to the increased deposit, which leads, in time, to an increase in the velocities and a decrease in the deposition. When breakthrough occurs, an insufficient depth of filter bed is available for removal and the run has to be terminated. The majority of authors, however, dismiss the initial stage as not being part of the average filtration run, thus taking into account only the three parts in which the transient stage is divided [18].

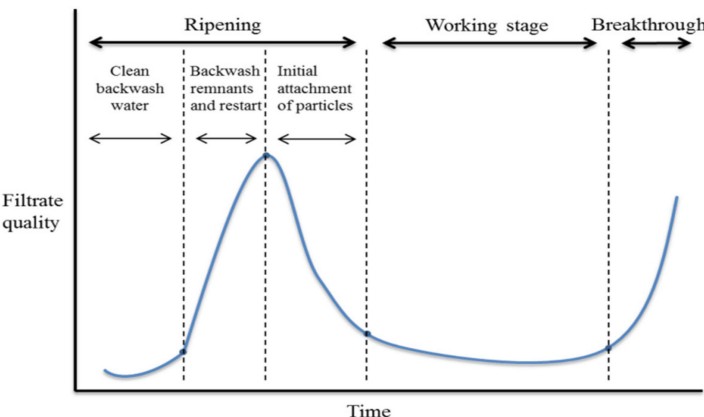

**Figure 2.** The filter cycle (reprepared using the data from [19]).

On the other hand, the macroscopic model does not take the transport/attachment problem into consideration. The overall removal is obtained via the integration of a mass balance relating the particle concentration to time and the kinetic equation relating the variation in concentration to the filter depth.

Transport in aqueous solutions involves multiple mechanisms (Figure 3) in a laminar flow. The occurrence of each mechanism depends upon the particle size. If the particle size is larger than the void size, straining is involved; for smaller particles sedimentation, interception and diffusion are dominant. The latter is more relevant for particles below 1 μm, while sedimentation and interception involve particles above 1 μm. Yao et al. [20], while modelling filter efficiency as a function of the particle size of contaminants, concluded that, for particles over 1 μm, the transport efficiency increases with the particle size, while for lower values it increases with decreasing particle size. A minimum is reached around 1 μm, a size that tends not to be removed during the run [17]. Among the protozoa *Giardia Lamblia* and *Cryptosporidium Parvum*, the former (10–15 μm size) is successfully removed by sedimentation. Cryptosporidium, ranging between 3 and 5 μm, is closer to the values of minimum transport efficiency [16,20].

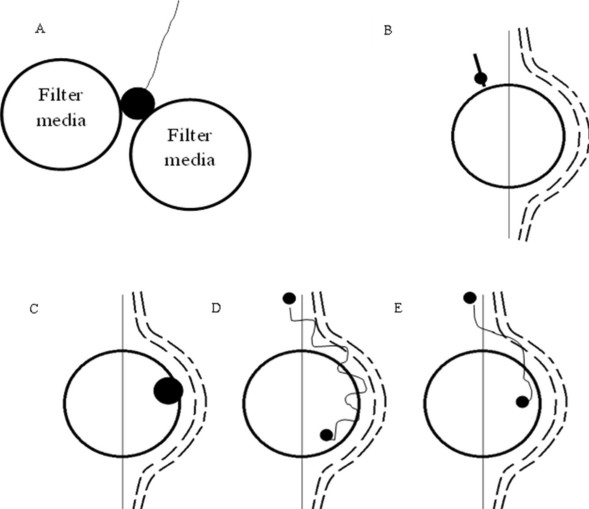

**Figure 3.** Transport mechanisms in water filtration: (**A**), straining; (**B**), sedimentation; (**C**), interception; (**D**), diffusion; (**E**), hydrodynamic (reprepared using the data from [10,12,20]).

The following mechanisms have been described in the literature:

1.  Straining. It is not desirable as collected particles clog the upper part of the bed (blinding), preventing an efficient use of the filter [12].
2.  Sedimentation. This is favoured when the density of the suspended material is greater than that of water. The particle will deviate from the streamline because of gravity and it will impact the medium surface [12,20]. This depends upon particle density and temperature [16], the diameter of the particle and more generally on the ratio between the settling velocity of the particle and the velocity of the fluid approaching the media [12]. Larger particles and lower filtration velocities will lead to higher collection efficiency for this mechanism [9].
3.  Interception. This occurs when a particle is transiting within a distance equal to its radius from the surface of the grain. The contact between the particle and the grain can result in attachment (12). The mechanism is very similar to straining, but smaller particles are involved [6,12,21]; it depends on the ratio of the particle diameter to the media diameter [12]. Its efficiency increases with increasing particle size and decreasing collector size [9].
4.  Diffusion. This is due to the thermal energy of the fluid, which is transferred to the particles. This causes them to drift from the streamlines to impact the surface of the grain or on other particles [9]. As mentioned previously, diffusion is efficient for sizes below 1 µm because viscous drag is not restricting the particles; the lower the particle size, the more significant the mechanism [12].
5.  Ives [12] adds inertia. The streamlines tend to diverge from the grains when approaching them, but particles with sufficient inertia might proceed unchanged and impact on the grains. It is, however, negligible for water filtration because of small mass and density differences [12,22].
6.  Furthermore, every particle is subjected to hydrodynamic action, caused by the velocity gradients within pore openings. As it experiences higher velocities on one side, the particle tends to rotate and create an additional spherical field, which causes the particle to move across the flow field. Because of deformable non-spherical shapes and non-ideal flow conditions, the results are non-predictable random paths, leading to movement across the streamlines and collision with the grains [12,23]. This is usually negligible; however, it appears to be more effective for lower particle–grain size ratios [11].

Flocculation within the filter pores might also occur, increasing the removal possibility [12]. This does not constitute a proper transport mechanism and only contributes in a small measure.

It is important to stress that the mechanisms act simultaneously, so that the effective transport of a particle cannot be ascribed to a single one, but to all of them. However, when considering water filtration, straining does not give a consistent contribution to the removal because of the size of the pores in comparison to the size of the particles [16]. Diffusion is not very significant in conventional rapid filtration, as the chemical pre-treatment favours the aggregation of smaller particles [9]. If biologically active filters—such as slow sand and granular activated carbon (GAC)—are considered, biodegradation has to be included among the mechanisms. Moreover, GAC filters enable the adsorption of certain contaminants [24].

As the particle approaches the collector hydrodynamic retardation occurs. This is due to the resistance caused by the displacement of fluid [15,16]. This depends on the distance between particle and grain surface, tending towards infinity as the distance tends to zero. Hydrodynamic retardation is more relevant for large, less dense flocs and particles than for the smaller and denser ones [9].

While transport is mainly a physical step, attachment is mostly chemical and it requires previous destabilization [17]. It is a consequence of short-range surface forces, namely van der Waals, electric double-layer repulsion and hydration forces, and the principles of these forces can be found in the Supplementary Materials.

Filter effluent particles can be divided among influent particles that are never deposited and particles that detach after deposition [25]. Detachment has been identified as the major cause of the presence of particles in the effluent. This does not occur until a specific value of the deposit is reached; afterwards, it is concurrent with attachment reaching an equilibrium [18]. An increase in the flow rate might cause the detachment of those particles that are less strongly linked [22]; it is generally agreed, though, that the main cause of detachment is the increase in velocity within the pores due to the deposits [26]. This can also be due to the impact of incoming particles on unstable deposits [14]. Detachment occurs when the hydrodynamic force becomes greater than the adhesive force and it acts via three mechanisms; rolling, sliding and lifting. As rolling does not necessarily lead to detachment, sliding and lifting are considered to be the main mechanisms [26].

## 2.2. Filtration Operating Setup

Filtration can be performed in two different ways: either as direct filtration or as a more conventional approach, where it is preceded by coagulation and clarification; this is the custom for waters containing a high level of particulate matter. Direct filtration is placed directly after flocculation, so that there is no separate clarification step. Direct filtration should be applied only when the average turbidity does not exceed 10 NTU, with peaks below 40 NTU and total organic carbon (TOC) not above 2 mg/L [5]. Moreover, media with a large grain diameter have to be employed in order to decrease the head loss [27]. Polymers are generally used before direct filtration [1], but if metal salts are chosen, small doses should be used to obtain small filterable flocs [5]. O'Melia [9] actually differentiates between contact (or in-line) filtration and direct filtration; in the former, the destabilized particles are directly sent to filtration, eliminating flocculation as well.

Additionally, filters can be classified in terms of continuous or semi-continuous operation, the latter being the case when the filter has to be put offline to be backwashed. The use of semi-continuous operation is the most common [28].

The process can be driven by gravitational force-rapid gravity filtration, or by pressure-pressure filtration. Both applications depend only upon physical removal [1]. The differences lie mainly in the pressure required, the filtration rate, and the type of vessel [29]. Pressure filters allow for significantly higher flow rates [8]. An additional type of filter; slow sand filtration, was historically the first one to be employed. It acts through a combination of straining and biological action [5]. Rapid sand filters were later favoured over slow sand because of the large area required by the latter to maintain a sufficient output and the excessively low filtration rates (0.1–0.3 m/h, or lower), also due to the smaller size of the sand grains [8]. Recently, it has been successfully adopted in many small communities because of the efficacy in removing protozoa pathogens [8,30]. Slow sand filtration is not, however, effective when

dealing with high levels of turbidity and algae because of a limited removal ability and long ripening times [5,31]. It also shows low efficiency in removing organic material and for this reason the use of granular media amendments (such as granulated activated carbon (GAC), anthracite, etc.) has been proposed. Resins and GAC are effective but lead to an excessive head loss; the other options have not proven to be more efficient than conventional operations [4,31]. Today, the majority of filters are of the gravity type and pressure filters are partially used in small plants. Rapid gravity filtration has the advantage of employing higher rates than slow sand filtration and it can be employed in either a single media or a multimedia configuration, coupling different materials together to increase the efficiency.

Filtration configurations affect the filtration performance. As single sand medium filters cannot always perform adequately to achieve the treatment tasks, a solution was found in the introduction of dual media filters by placing a denser material at the bottom and a lighter one at the top, with decreasing size. In the most common configuration, a layer of anthracite is placed on top of a sand layer; in some cases, an additional layer of garnet is added [1,5]. The filtrate quality is comparable to that of sand, but filter runs can be up to 1.5–3-times longer with similar filtration rates. Attempts have also been made towards an increase in the filtration rate with dual configurations, though this is often accompanied by the use of coagulation aids. The size of the flocs, resulting from coagulation, is important when employing dual configurations. If the flocs are too small, they might bypass the first layer and lead to a fast clogging of the sand layer. If too large, the anthracite layer would be quickly clogged [5]. The effective size for anthracite is generally 1.5 mm, though varied values are obtained in different parts of the world. The depth of the anthracite layer is usually set around 150–300 mm, while for sand it is at 450–600 mm [32]; the opposite proportion is given by Binnie and Kimber [1].

Zouboulis [27] compared the performance of a single medium sand filter and a dual media sand/anthracite filter for conventional and direct filtration. During conventional filtration, the dual media configuration was able to operate for a longer cycle, leading to a 10% higher water production; the length of a dual media cycle was 2–3 times that of the single medium, with final head loss values being less than half (Figure 4a). For turbidity removal, both configurations showed values well below 0.2 NTU, with the dual media configuration performing marginally worse (Figure 4b). Direct filtration was more difficult to control. For small doses of coagulant, the single medium did not reach the required level of turbidity, presenting values of 0.5–1 NTU. The dual media filter was more effective, achieving turbidity values slightly higher than those with conventional operations (0.2–0.3 NTU) [27].

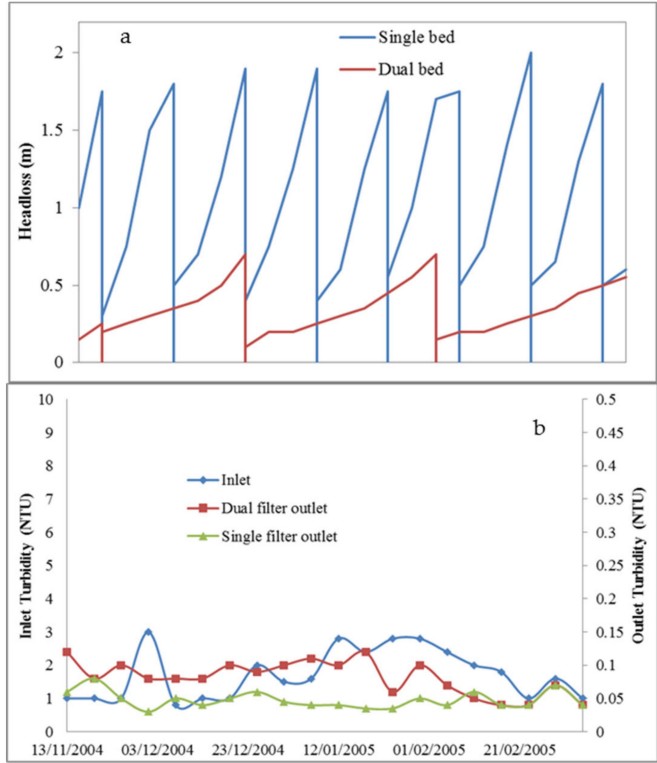

**Figure 4.** (**a**) Head loss development (19/02/2005–25/02/2005) and (**b**) values of turbidity for conventional treatment (reprepared using the data from [27]).

In order to adopt higher filtration rates, there has been an attempt to use anthracite in coarser mono-medium filters, with a deeper bed (1.8 m); though this route was soon abandoned [30]. The main issue with anthracite is its higher cost and the limited number of sources worldwide; for these reasons it has occasionally be replaced by high grade bituminous coal [5]. Other materials have been used in specific circumstances, such as GAC in place of anthracite or sand for the removal of odours. It is more common, however, to include an adsorption stage after the filter with a longer contact time that increases the efficacy of the GAC [30].

### 2.3. Process Performance Monitoring and Filter Backwash

Filtration performance can be monitored by regularly checking the flow rate, head loss development and effluent quality characteristics (e.g., turbidity). Flow rates vary depending upon the type of filter, the filtration media, the plant and the clarified water quality. Typical rates for deep coarse beds are around 6–8 m/h, though they can be increased up to 12–15 m/h if dual media filters are employed and the feed water shows consistent good quality [1]. The value 6–7 m/h should not be exceeded where there is a presence of oocysts in the raw water; for graded sand beds the rates applied should be 25% lower. Rapid changes in the rates are to be avoided, as they can lead to excessive shear on the deposits, with release of particles into the effluent [19]. It is observed that, at rates above 15 m/h, the quality of the water deteriorates, and above 20 m/h, the head loss development becomes too rapid [32]. Higher filtration rates imply a faster development of the head loss, but they give higher productivity, so that the ideal balance has to be found [14]. Clean bed head loss is the resistance initially encountered by the flow through the bed. The head loss development is linear if the entire depth of the bed is used, but exponential if only a small depth near the surface is used [23]. During operation it is important to avoid negative head, which is to say pressures below atmospheric pressure within the bed. This occurs when the head loss exceeds the static head (water depth) at any given depth [14]. Depending upon the media, this could occur at different points in the bed. The consequences might

include the release of dissolved gases, poor effluent quality, premature ending of the run, cracking and mud ball formation [5,6].

Once the rate has been chosen, it can be applied in constant or declining mode. With declining rates, the amount of water fed to the filter is maintained throughout the run. Initially, when the filter is clean, a low head loss (0.2–0.3 m) is measured and the maximum flow is limited via a valve at the outlet. During the run the head loss increases and the flow rate diminishes, until the minimum acceptable flow rate is reached, or backwashing is scheduled. The main concern over this mode of operation is related to the negative effect of the high filtration rates at the start. Constant rate filtration is achieved by controlling the flow either at the inlet or the outlet. The head loss through the filter increases during the run but the flow remains unchanged. Backwashing is performed either when the terminal head loss or turbidity values are reached, or after a set operation time, generally 24–60 h [1,5,12]. It is desirable for the limiting head loss (1.5–2 m) to occur at the same time as the breakthrough, so as not to lose filtering capacity [12]. The safety of operations is, however, more important and it is preferable not to wait until the filters are close to breakthrough.

Figure 5 shows the filtrate turbidity during a typical rapid sand filter run. The filter is commonly allowed to reach a head loss of 2 m before being backwashed. The employment of rapid sand filters gives rise to a specific issue; during backwashing smaller particles would settle closer to the top while larger ones would reach the bottom of the filter, leading to stratification of the bed. The top, finer layer would retain the majority of incoming particles, with an ineffective use of the depth and shorter runs.

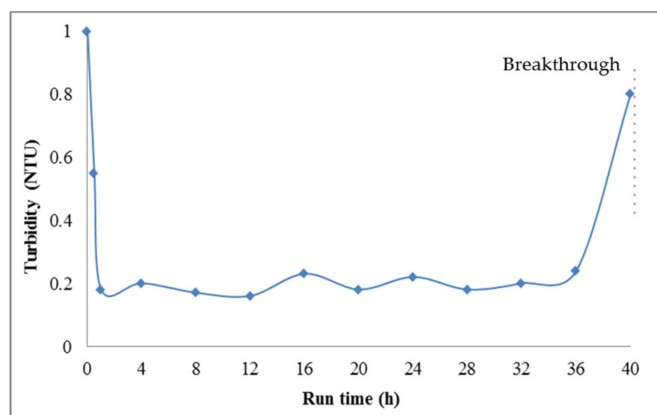

**Figure 5.** Turbidity as a function of run time for a typical filter run (reprepared using the data from [32]).

The backwash is performed by inverting clean water so that the filter is flushed. The rates applied are generally above the minimum fluidisation rate—i.e., the rate at which the drag forces overcome the weight of the grain and the bed starts to expand (be fluidised) [1]. This is affected by features of the media, such as size and density and by the packing of the bed; the lower the voidage, the lower the minimum fluidisation rate [33]. Voidage refers to the free space within the filter bed, which is available for the retention of solids [34]. There is also a significant temperature effect due to the influence of the viscosity of the water [6]. Employing the right backwashing rate is fundamental, as rapid flow might lead to loss of media and to an excessive consumption of water. Indeed, a variable amount of filtered water and the necessary pressure have to be provided; water consumption is usually around 2.5 times the bed volume [5]. Ineffective washing is the consequence of low backwash rates [6,33,35] together with floc build-up, which can lead to mud balls, jetting, cracks or shrinkage within the bed [5].

It is common to use air and water simultaneously, which has been demonstrated to be the most effective approach for backwashing, although the use of air scouring requires particular care as it could lead to modifications in the structure of the filter [33] and it increases the chances of media loss [19]. A backwashing procedure is described by Ratnayaka et al. [5] for a deep bed homogeneous sand filter with air (3–4 min) and water at a flow sufficient to achieve fluidisation (4–6 min), which is suggested as a practice suited to all rapid gravity filters. Full fluidisation of the bed at the end of the backwash

is required if multimedia filters are being cleaned, as it allows proper re-stratification. In practice, intermixing at the interface occurs in most cases for a depth of 100–150 mm. It has been argued that this intermixing, by reducing the voidage of the coarser media, increases the filtrate quality and leads to higher head loss development [5]. The intermixing should not, however, occur for more than 20% of the layer [36], or a zone with extremely low permeability might form with risk of clogging [37].

According to some sources, rates should be around 10–30 m/h for the water flow, while airflow rates can be between 50–80 m/h [1]; others recommend a value between 23 and 58 m/h for the air flow, and 7–25 m/h for the wash with water [32]. During the first few backwashes, it is likely that a certain number of fine grains will be removed from the bed. This might affect the packing of the bed, so that adjustments to the process and the flow rates are required [33]. A solution to avoid negative effects is the addition of 30 to 50 mm of media to the bed depth [29]. Media losses are more common for anthracite and GAC, due to their lower attrition resistance [38]. If the issue is prolonged after the first few cycles, this might signify that the rates applied are too high, or that the filter does not possess the required characteristics [4].

One of the main concerns related to backwashing is the high level of turbidity after the filter has been returned to service. This phase is referred to as ripening (Figure 2). This is due to the removal of particles that contribute to filtration and to flocs not properly discharged during the last backwashing phases [32]. Three stages have been identified; initially, the filtrate is affected by the remaining backwash water located in the underdrains of the filter and subsequently by contaminants left above and within the bed.

Finally, the effectiveness of filtration is reduced because of the lack of the additional collection capability due to retained particles [39]. This issue has received increasing attention as it has been shown that the transmission of *Giardia* and *Cryptosporidium* cysts can occur during this phase [16]. Different techniques are available to minimize the issue; the addition of a coagulant to the backwash water or to the influent as the filter is returned to service, implementation of a terminal sub-fluidised rinse, filter resting (delayed start), discard of the first fraction of effluent (filter-to-waste) and variation of filter rates (starting from low rates and gradually increasing; called slow start) [5,24]. It is considered normal for a filter to produce effluent with a turbidity of 0.5–1 NTU after being returned to service, though it should then decrease to 0.2 or less in the first 30 min of operation and to 0.1 after an additional hour [1,8]. Hess [19] reports the objective of 0.3 NTU after backwashing and below 0.1 NTU within 15 min of the return to service.

## 3. Development and Testing of New Filtration Media

Traditional filter media are sand and anthracite which have been and are used in a variety of filtration practices in the last few decades. The development of new media has been researched and applied. In some cases, the focus was on naturally available materials, such as pumice [40] or crushed quartz [41]. In others, natural materials were improved through the application of coatings [42] or via physical processes (e.g., expanded aluminosilicate/clay). The latter, in particular, has been successfully employed for drinking and waste water filtration, and so is an interesting candidate. Diatomaceous earths have been considered, as they do not require pretreatment with coagulants. However, a leakage of material into the effluent is likely [4]. Numerous other approaches involving granular materials have been investigated. Farizoglu et al. [40] combined the use of an upward configuration with pumice, a light and porous rock; this reduces head loss, increasing the length of the filter run. Experiments have been carried out only without the addition of coagulants [40]. A similar material, natural pozzolan, has also been tested for use in upward direct filtration with low flow rates. The study highlights the danger of the transportation of particles outside the column during backwashing and the necessity of slightly acidic pH (4) to obtain consistent efficiency [43]. Iron oxide coated pumice has been employed successfully for the removal of DBP precursors; the coating results in a material with a high surface area and stability at the pH of natural waters [42]. Suthaker et al. [41] employed crushed quartz in a pilot-scale study to improve particle and turbidity removal. It was found

that it gave better performances than anthracite/sand, though the latter, as a dual media configuration, allowed longer filter runs [41]. As well as the stated alternative media, in subsequent sub-sessions, we will mainly discuss and compare the filtration performance of sand and anthracite with that of expanded aluminosilicate, recycled glass media, polypropylene fibre and sand with granular activated carbon. These materials have proven effectiveness and compliance with the current legislation for drinking water treatment (e.g., Drinking Water Inspectorate 2016 [44]).

### 3.1. Expanded Aluminosilicate–Filtralite

Filtralite is produced in Norway from expanded clay aggregates obtained by firing at 1200 °C, followed by crushing and grading. It is available in a variety of grain sizes and densities—from NC, the lightest, to MC and HC, the heaviest. Mitrouli et al. [45] compared the effectiveness of anthracite and Filtralite MC (1.5–2.5 mm) as part of a dual media configuration, including sand, for the pretreatment of seawater. The Filtralite/sand filter showed equivalent or improved turbidity removal, residual particles and TOC removal for various filtration rates. The size of sand was almost twice that of the 16/30 (sieve identification) specification normally used for drinking water treatment; the medium is likely to have underperformed in the circumstances. Indeed, smaller grain sizes provide a higher removal efficiency as the particles have to travel shorter distances before attachment occurs. Similar grain sizes should be used for a reliable comparison. Table 1 shows comparative operating parameters by the researchers that assessed filtration performance of various Filtralites and sand.

**Table 1.** Comparison of relevant parameters for studies on Filtralite.

| Flow Rate (m/h)/Type of Test Water | Bed Depth (cm) Filter 1 | Size Ranges (mm) Filter 1 | Bed Depth (cm) Filter 2 | Size Ranges (mm) Filter 2 | Reference |
|---|---|---|---|---|---|
| 5, 10, 15/Raw seawater | Anthracite: 70 Sand: 50 | 1.2–2.5 0.8–1.25 | Filtralite MC: 70 Sand: 50 | 1.5–2.5 0.8–1.25 | [45] |
| 5, 10, 15/Raw seawater | Anthracite: 70 Sand: 50 | 1.2–2.5 0.8–1.25 | Filtralite NC: 70 Filtralite HC: 50 | 1.5–2.5 0.8–1.6 | [46] |
| /Tap water with added humic concentrate and/or bentonite clay | Anthracite: 60 Sand: 35 | 0.8–1.6 0.4–0.8 | Filtralite NC: 48 Filtralite HC: 47 | 1.5–2.5 0.8–1.6 | [47] |
| 10/Raw water | Anthracite: 50 Sand: 50 | 1.7–2.5 0.6–1.18 | Filtralite NC: 50 Filtralite HC: 50 | 1.5–2.5 0.8–1.6 | [48] |
| 8.6, 11.1, 13.6/Clarified water | Sand: 60 | 0.59 ($d_{10}$) | Filtralite: 60 | 0.77 ($d_{10}$) | [49,50] |
| 5–12/Tap water with added humic concentrate | Anthracite: 60 Sand: 35 | 0.8–1.6 0.4–0.8 | Filtralite NC: 60 Sand: 35 | 0.8–1.6 0.4–0.8 | [51] |

The major advantage given by the use of Filtralite was, however, the slower head loss development which could lead to longer run times [45]. Filtralite presents a rough surface characterised by a high number of pores and crevices; the texture is considerably different from anthracite. This allows slower head loss development, but at the same time causes higher bed porosity; meaning that the particles will have to be transported for longer distances to reach the grains. An attachment is thus less likely to occur [46]. The lower retention of material contributes to the slower head loss development, as the amount of accumulated deposits is lower [49]. A second investigation from the same authors involved the use of a Filtralite NC 1.5–2.5 mm and HC 0.8–1.6 mm (Mono Multi) dual configuration in comparison with sand/anthracite. The two showed similar performance with regard to turbidity, and once again the behaviour differed in terms of head loss development [46]. The Filtralite NC/Filtralite HC configuration is also the basis for Figure 6, produced from the manufacturer's own testing, which shows both the head loss development and the expansion of the bed during backwashing as a function of the flow rate.

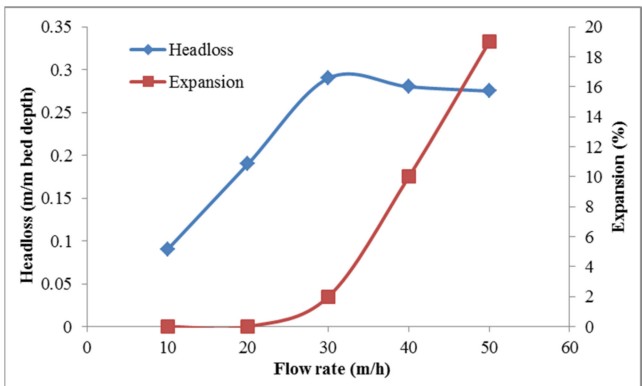

**Figure 6.** Backwash head loss and expansion as a function of flow rate (reprepared using the data from [49]).

A similar analysis was performed by Saltnes et al. [47] on raw water with high humic content, treated with a variety of coagulants. Once again, the Mono Multi configuration is compared with sand/anthracite. The removal of TOC is similar; when employing metal-based coagulants, the coarser grains of Filtralite were not able to reduce turbidity and residual metal sufficiently. The authors suggest employing a deeper bed to obtain a removal comparable to the traditional setup [47]. Mikol et al. [48] performed the same comparison with similar results for roughing filters, which are used before slow sand filters or membranes. In addition, the backwashing of the configurations was investigated, with the conclusion that no major changes in the process would be required [48].

Davies and Wheatley [50] tested Filtralite as a single media in pilot scale trials. According to the results, as a single media, it tends to show worse turbidity removal performances (still below 0.1 NTU) but slower head loss development than the traditional setup. When tested in a laboratory setting, Filtralite appeared to have even lower effectiveness, achieving 70% turbidity removal against 88% for sand, even at slow flow rates [46]. Thus, it appears to be more effective as part of a dual media configuration or for the treatment of low turbidity raw waters [50]. NOM removal was targeted by Eikebrokk and Saltnes [51] in trials employing Filtralite and several coagulants. The differences between the Filtralite/sand and anthracite/sand configurations were small in terms of effluent quality; the same advantages for head loss (12–28% lower rate) and run length were measured as highlighted by the other authors.

*3.2. Glass-Based Media*

Recycled glass has been investigated as a possible replacement for sand. Ratnayaka et al. [5] mention it as a suitable substitute because of its similar specific gravity. Research on such filter media usage has been reported; mainly, they have been used in wastewater treatment [52,53], swimming pool water cleaning [54], and drinking water treatment [55]. With regard to drinking water treatment, recycled glass has shown improved performance compared to sand. A report from the Water Research Centre relates that full-scale applications of glass media could lead to 50% longer filter runs with a similar water quality [55]. They maintain that the media has a life comparable to that of sand and that it presents lower friability during the filtration-backwashing cycles [56]. Davies and Wheatley [50], though, showed lower mechanical durability of the glass media compared to sand; this could lead to issues towards the end of the filter life or to untimely media changes [49]. Glass also shows a smoother surface when compared to sand; more efficient backwashing could be performed with lower costs, as the detachment of particles is more easily achieved [49].

Several academic studies have been conducted, both for gravity filtration and pressure filtration, in single and dual media configurations. Single media configurations included sand or glass with or without gravels, while for the dual media there was an additional layer of anthracite. The glass used was, in most cases, a non-specifically recycled glass. The details regarding the setup and grain characteristics can be seen in Table 2. Both for sand and glass media, materials with different properties

were used in the different studies. This is particularly clear when considering the effective size for sand, which varies between 0.33 and 0.97 mm. A similar difference is displayed for the uniformity coefficients (1.27–1.82), while it is less pronounced for glass-based media (0.59–0.98 mm, 1.21–1.58). Smaller values of effective size generally indicate a material with good removal efficiency, leading to a fast head loss development. A slower head loss development and slightly lower effectiveness can be expected when higher values of the property are measured. The variability will thus affect the comparison between the media. Indeed, where the physical properties of sand and glass differ, discrepancies in terms of efficiency can also be ascribed to the properties and not only to the material itself. In order to obtain comparable results, similar sizes and uniformity coefficients should be employed [57].

**Table 2.** Comparison of the relevant parameters for studies on recycled glass media.

| Type of Configuration/Type of Test Water | Coagulant | Bed Depth (cm) | Flow Rate (m/h) | Effective Size Glass ($d_{10}$, mm) | Effective Size Sand ($d_{10}$, mm) | Uniformity Coefficient Glass (UC) | Uniformity Coefficient Sand (UC) | Ref |
|---|---|---|---|---|---|---|---|---|
| Dual media/Raw water | PACl | Anthracite: 60 Sand or Glass: 40 Garnet: 6 | 5 | 0.59 | 0.33 | 1.58 | 1.82 | [59] |
| Single media/Raw water | Alum (plus additional filter aid) | Sand or Glass: 90 Gravel: 10 | 7.5, 10, 12.5 | 0.98 | 0.97 | 1.31 | 1.27 | [60] |
| Single media/Raw water | Alum or Ferric Chloride | Sand or Glass: 104 | 11.5 | 0.77 | 0.79 | 1.41 | 1.33 | [58] |
| Dual media/Raw water | Alum or Ferric Chloride | Anthracite: 41.5 Sand or Glass: 62.5 | 11.5 | 0.77 | 0.79 | 1.41 | 1.33 | [57] |
| Single media/Raw water | Ferric sulphate | Sand or Glass: 60 | 0–9 | 0.76 | 0.59 | 1.21 | 1.27 | [50] |
| Single media/Tap water with added kaolin clay | No coagulation | Sand or Glass: 60 Gravel: 41 | 8.6, 11.1, 13.5 | 0.76 | 0.59 | 1.21 | 1.27 | [49] |
| Single media/Raw water | PACl | Glass: 80 | 6 | 0.56 | 0.58 | 1.28 | / | [55] |

Recycled glass media were tested with anthracite as a dual media configuration against sand/anthracite. The performance was evaluated mainly through particle counting. It is interesting to note that a ripening period of 15–20 min was recorded for glass/anthracite, after which the effluent quality was consistently around 50–70 particles/mL (for particles >2 µm). Sand/anthracite required around 10 min, presenting afterwards 25–50 particles/mL (Figure 7). The number of influent particles, however, varied between the runs; the average for the glass was 500–1000 particles/mL, while for the sand filter it was 1000–1500.

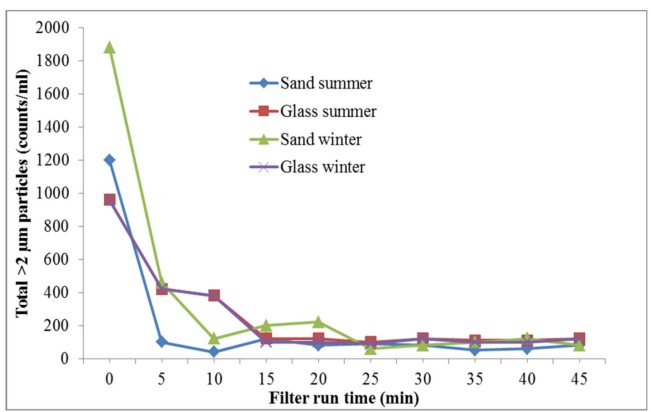

**Figure 7.** Ripening times in winter and summer conditions (reprepared using the data from [57]).

Soyer et al. [57,58] looked at both single and dual media configurations for gravity filtration in two different studies. The recycled glass used in the study was once again not specifically manufactured for water filtration purposes. The single media filters showed comparable removal efficiencies for both turbidity and particle count; the head loss was lower for glass (up to 50%). Fluidisation experiments led to similar expansions for the glass and the sand filters, with small differences at higher backwash rates [58]. Similar observations were obtained with dual media configurations; a slightly increased

effectiveness towards turbidity was measured for glass/anthracite. The head loss was on average lower for glass/anthracite, though the intermixing at the interface was higher. This did not have a strong overall influence and the effect was still that of a slower development and a lower clean bed value [57].

Recent research by authors [61] demonstrated that the glass media configurations granted similar performance in relation to that of sand/anthracite in terms of the removal of particles and organic contents. However, a slower head loss development with glass media configurations (both Filtec/Filtralite and AFM/Filtralite) was observed [60]; final head loss values of glass media were more than 60% lower than those obtained by sand/anthracite (Figure 8). Moreover, a long filtration run would lead to a lower frequency of backwashing, in turn leading to the reductions in energy and backwashing water consumption. This, combined with the less particle breakthrough witnessed in the study, could lead to a promising application of glass media in the water industry.

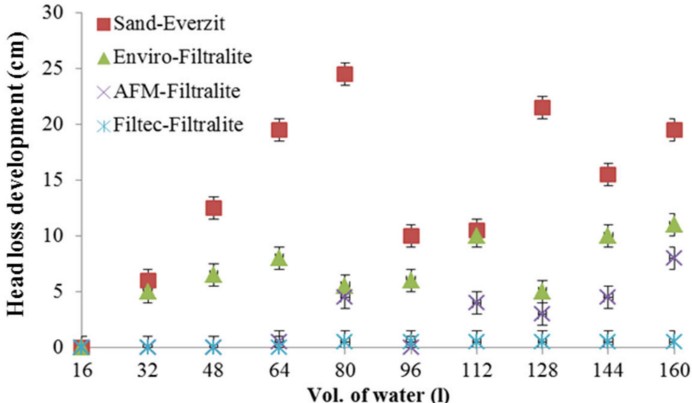

**Figure 8.** Head loss development in four dual media configurations for the entire run with poly-aluminium chloride as coagulant (reprepared using the data from [61]).

When comparing glass media with sand in water filtration performance, one interesting issue is raised; if glass media could also support biological activities? This is because that rapid sand filters can be managed to promote biological activity, particular when prechlorination has been replaced by preozonation. One example of such studies has demonstrated that sand and anthracite media in rapid gravity filters support biofilms development and biological activities if prechlorination is not applied [62]. Biological activity in both slow sand filtration and rapid sand filters is attributed to the development of a complex biofilm coating on, and sometimes bridging between, sand grains. The surface properties of media are one of critical factors to affect their biological activity. For drinking water treatment, microorganisms are concentrated in areas that are sheltered from fluid share forces. Surface irregularities of sand or media would provide a sheltered environment for biofilms from fluid share forces, and consequently biological activities.

Table 2 shows the effective size ($d_{10}$) and uniformity coefficient (UC) of sand and glass and, in most cases, $d_{10}$ and UC of sand and glass are close. The research conducted by authors demonstrates that the glass media have higher angularity (Figure 9a) but sand appears more rounded or higher sphericity (Figure 9b). it is expected that the porosity of the filter containing glass media would be slightly higher than the porosity of the sand filter because of greater angularity of glass media [63]. This was confirmed in a study in which the porosity for a sand media filter was 0.47 vs. the porosity of 0.52 for a glass media filter [59]. An exploration of biological activities of glass media in slow sand filters was reported. The glass product from the pulverizer meets the USEPA specifications for slow sand filters. Over an 8 month period of continuous operation, the performance of the glass media was as good or better than the silica sands, with removals of 56% to 96% for turbidity; 99.78% to 100.0% for coliform bacteria; 99.995% to 99.997% for giardia cysts; 99.92% and 99.97% for cryptosporidium

oocysts. Based on the results, pulverized glass is an effective alternative to silica sand as a filter media for slow sand filtration [64].

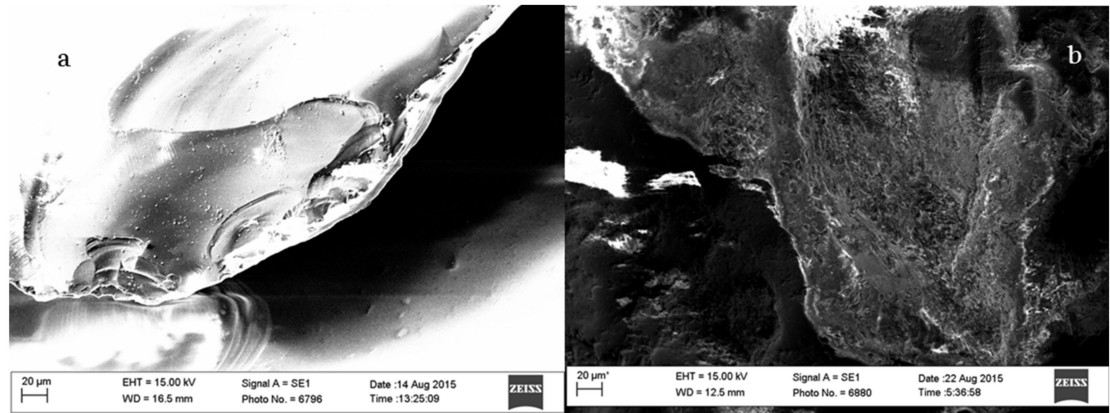

**Figure 9.** SEM image of (**a**) Enviro Glasmedia and (**b**) sand (from authors' unpublished work).

### 3.3. Polypropylene Fibre

Flexible fibre filters are recently developed and applied to wastewater and water treatments, and polypropylene fibre is one of commonly used polyamide fibres for constructing filtration modules. Polypropylene fibres were obtained from ropes sold for agricultural purposes and characterized in order to evaluate their use as filter media [65]. Through SEM analysis (see Figure 10), the researchers observed that polypropylene fibres are considerably uniform, similar to polyamide fibres, small diameters (about 34 µm) of polypropylene fibres might be associated with higher filtration surfaces. On the other hand, residues were observed on the surface of fibres used as filtering media for 3 months. This showed the effectiveness of the filtration process, with the attachment of solids on the fibres' surfaces.

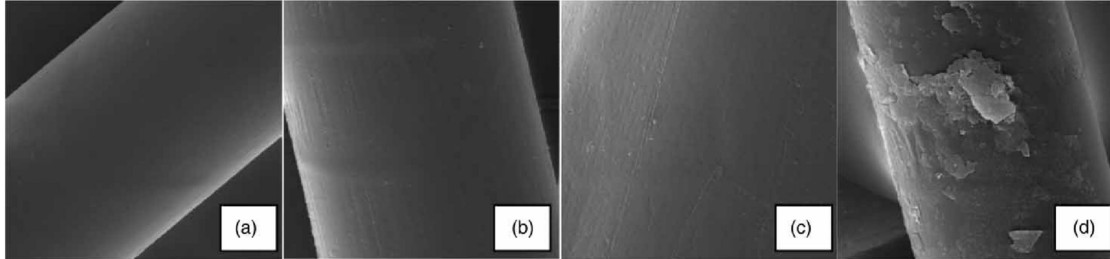

**Figure 10.** SEM images with magnification of 2000: (**a**) without any pre-treatment or use as filter media, (**b**) after solubility assay in HCl, (**c**) after solubility assay in NaOH, (**d**) after use as filtering material for 3 months [65].

The results of this study suggest that flexible fibre filters built with polypropylene fibres can produce high-quality water. Colour values were always below 1 CU for all filters during the filter run (before turbidity and head-loss limits were reached), and the filter can run at 20–80 $m^3/m^2$ h to achieve residual turbidity less than 0.5–1 NTU depending on the filter's depth. Moreover, polypropylene filter backwashing water demand was about 2%, suggesting high water productivity. Finally the study found that the use of different fibres and coagulants in the construction and operation of this type of filter did not affect its performance, suggesting that other materials could also be used alternatively, in order to include available and/or natural resources.

### 3.4. Sand with Granular Activated Carbon

For the removal of organic contaminants from water resources, granular activated carbon (GAC) is used to replace the granular media commonly used in rapid filters (sand or sand and anthracite)

or as a step after conventional filtration due to its high surface area, porosity and surface affinity, which provide a higher capacity for the adsorption of organic molecules. Then, dual media filters with sand and GAC have been researched to remove both particles and organic contaminants [66,67]. A recent work [68] examined the effect of applying dual media configured filtration to the reduction in phenols in a conventional drinking water treatment plant, using two types of GAC (vegetable and mineral) and three GAC/Sand configurations (1/0; 0/1; 0.5/0.5). The configurations of filters can be seen in Table 3.

**Table 3.** Configurations used for the evaluation of sand-GAC filtration. (reprepared using the data from [68]).

| Configuration | Vegetable Activated Carbon (VAC) (%) | Mineral Activated Carbon (MAC) (%) | Sand (%) |
|---|---|---|---|
| C1 | 100 | - | - |
| C2 | - | 100 | - |
| C3 | - | - | 100 |
| C4 | 50 | - | 50 |
| C5 | - | 50 | 50 |

The study reveals that the dual filters with sand and GAC ensured better removal of TOC and phenols. As shown in Figure 11, with the GAC filters, the maximum removal efficiencies of 80% for TOC and 99% for phenols were achieved by either mineral activated carbon (MAC) (C2) or sand with MAC (C5), while with sand alone (C3), the maximum removal efficiencies were 26% for TOC and 10% for phenols. In general, it was noted that mineral GAC (configurations C2 and C5) was more efficient than vegetal GAC (configurations C1 and C4). However, the adsorption efficiency of GAC depends on many other factors, such as the water quality characteristics, contact time, filter depth and the volume of the adsorbent, etc., and the operating conditions must be evaluated in each case to reach the highest treatment performance.

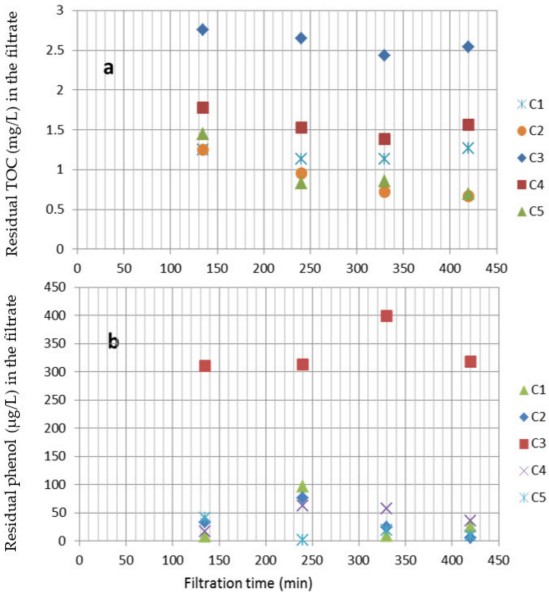

**Figure 11.** Residual concentration of contaminants in the filtrate over time for all filter configurations. (**a**) TOC, raw water TOC concentration = 3.369 mg/L; (**b**) phenol, raw water phenol concentration = 245.3 µg/L (reprepared using the data from [68]).

## 4. Conclusions and Future Work Recommendation

Filtration is one of the key processes in water treatment. More stringent regulations drive water industries to seek higher efficient filtration in cooperation with other processes. The effectiveness of alternative filter media and the benefits have been demonstrated in the published literatures including high filtration performance in removing residual particles and turbidity, minor modification requirement to the existing filtration configuration and slow head loss development, which results in potential long filter run period. Recycled glass and traditional sand media showed similarities for several of the properties; the replacement of the sand with the glass media would thus involve minor modifications to the existing configurations. Application of polypropylene fibres as filter media can produce high-quality water with low colour and turbidity (<1 CU for colour and <0.5 NTU for turbidity) and backwashing water demand was 2% only. The studies also reveal that the dual media configurations with sand and GAC significantly improved the removal of organic contaminants in waste water treatment.

Four categories of possible future work are suggested through this review, namely, full scale tests, variation of the operating conditions, expansion of the number of media tested and pathogen removal related analysis.

Evaluation of the performance of the alternative media should be conducted through full scale trials. The employment of a full-scale setup would allow for the exploration of several issues, such as the prolonged ripening presented by some of glass media. Full-scale backwashing tests would also be needed to confirm the results obtained via laboratory trials. Moreover, the total costs encountered in the application of the alternative media should be more precisely obtained via the full-scale trials.

In running full scale trials, various operating conditions should be tested according to the water treatment requirement, such as filter depth, column size and pre-treatment setup.

Extending the range of media tested could constitute to an additional alternative. For those whose regulations are not yet approved, a case could be made leading to their approval and application.

Finally, before real applications of the alternative media are implemented, the pathogen removal should be considered to test. This could be achieved by using actual raw water or by using a surrogate (e.g., inactivated oocysts) successfully employed in previous studies.

**Supplementary Materials:** The additional data and information are available online at http://www.mdpi.com/2073-4441/12/12/3377/s1, including Session 1, legislation and standards for safe drinking water; Session 2, the double layer model and hydration forces acting onto particle surface; and Session 3, development of mathematical models for the bed media filters.

**Author Contributions:** A.C. and J.-Q.J. have contributions to the conception and design of the work, analysis of the data, and drafting the work. J.-Q.J. gained the research funding, supervised the work and provided extensive contributions to the preparation of the final version of the manuscript. All authors have read and agreed to the published version of the manuscript.

**Funding:** This research was funded by Glasgow Caledonian University and Scottish Water.

**Acknowledgments:** The authors are grateful to Glasgow Caledonian University and Scottish Water for offering a studentship to Anna Cescon for the PhD study. The views expressed in this review are not necessarily representing those from the company.

**Conflicts of Interest:** The authors declare no conflict of interest.

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
