# Peer review of "Filtration Process and Alternative Filter Media Material in Water Treatment"

_water, doi:10.3390/w12123377_

Round 1

Reviewer 1 Report

I enjoyed reading your paper, which has been clearly written and contains interesting information on the application of (sand) filters. I really gives a nice and broad overview of the knowledge about filters and filter media.

I only have one question. Glass may be an interesting alternative for e.g. sand, partly due to its smoother surface. However, we have seen that biodegradation or organic micropollutants not only occurs in slow sand filtration, but sometimes also in rapid sand filtration. Do you think glass beads would also host micro organisms, who can degrade micropollutants? This may be an important aspect for some filter applications.

Author Response

We thank two reviewers’ constructive comments and amend the manuscripts accordingly. Changes have been made which can be viewed in the “revised manuscript-changes tracked”. Below are our responses to the comments.

It is indeed that rapid sand filters can be managed to promote biological activity, particular when pre-chlorination has been replaced by pre-ozonation. To address the above comment, we have added paragraphs with our work (SEM images of glass media and sand) to detail this interesting topic in session 3.2 (pages 16-17).

Reviewer 2 Report

This paper is an extensive litterature review of the deep bed media filtration (single , dual media). The theoretical part of the paper and the supplement lacks the full mention of important filtration models  mainly based on particle size (PSD), particle volume (PVD), attachment and detachment forces which gives more microscopic study possibilities and not only macroscopic analysis based on hydraulic head loss and turbidity or PSD removal only.

  • Vigneswaran S., and Chang J.S. (1986). “Mathemathical Modelling of the Entire Cycle of Deep Bed Filtration”. Wat. Air Soil Pollut. , 29, 155-164.
  • Vigneswaran S., Jing Song Chang, “Experimental Testing of Mathematical Models Describing the Entire Cycle of Filtration”, Wat. Res.:23,11,1413-1421.(1989).
  • Adin A., Rebhun M.,. (1977). A Model to Predict Concentration and Head-loss Profiles in Filtration”, J. AWWA. 8:444-451
  • Cikurel H., Adin A., Amirtharajah A., Rebhun M., (1996). “Wastewater Effluent Reuse by In-line Flocculation Filtration Process”. Wat. Sci. Techn. Vol. 33, No. 10-11, 203-211.
  • The last one is a paper showing the application of these theories on wastewater treatment. An attached paper is also emphasizing the way different coagulants and also media (depending on their shape factor and size)can be compared using these models.

This is important when taking into account bacterial and parsite and even some virus removal efficiencies. 

The media types studied were (either as single media or dual media): Sand, anthracite, GAC, Filtralite, AFM and polypropylen fiber. Mainly, sand/anthracite and sand /GAC media has been intensively studied both for water and wastewater treatment, so more references emphasizing which types of water were treated could be mentioned (to add to the Tables 2 and 3 which types of water were studied - synthetic, drinking water, well water, wastewater..).

This is important in view of the extensive treated wastewater reuse for unrestricted irrigation or even direct or indirect potable reuse (filtration being the preliminary step of a series of post treatment) more references can be added.

Author Response

We thank two reviewers’ constructive comments and amend the manuscripts accordingly. Changes have been made which can be viewed in the “revised manuscript-changes tracked”. Below are our responses to the comments.

Mathematical models have been developed to predict granulated media filter’s performance; we agree this is one of important aspects in the fields of filtration for water treatment. The subject itself should constitute to an independent review and the current manuscript focuses on the alternative filter media development and application. However, to address the comment by Reviewer1, we have added one session into “supplementary materials” to brief the media filtration models and list Reviewer2’s suggested literatures.

We also have added additional information on the type of water used for the results shown in Tables 2 and 3.

Round 2

Reviewer 2 Report

One comment for the authors' claim that "although the development of mathematical models is one of important aspects in the fields of filtration for water and waste water treatment, the current manuscript focuses on the alternative filter media development and application", is that:

Filtration is a process that has to be developed mainly experimentally (including alternative media development, biofiltration or flocculation processes).This has to be done at pilot scale to obtain a successful result due to multiple parameters that affect the process, making it more complicated, mainly in wastewater reuse technologies.

Mathematical models have laregly contributed, together with the field pilot work, to the development of the deep bed filtration and its application in the filtration of different types of water.

As mentioned also by the authors, lately more biofiltration processes (mainly activated carbon, antracite and sand) coupled with ozonation are being used in water reuse technologies. These processes are  effective to treat mainly non-biodegradable micropollutants, excess ammonia removal from secondary effluents by nitrification and reduce DBPs. The authors emphasize that the glass media can be a good replacement for sand media but mainly in slow sand biofiltration.

At higher filtration velocities (5-10 m/h) coupled with flocculation,sand-anthracite media has proven to be effective removing turbidity, particles, crypto, giardia, and if coupled with ozonation also has removed some micropollutants and reduce ammonia.

May be a parallel experiment with glass media can be performed in the future. A reference can be found at:

Zucker I., Mamane H., Cikurel H., Jekel M., Hübner U., Avisar D., (2015) "A Hybrid Process of Bio-filtration of Secondary Effluent followed by Ozonation and Short Soil Aquifer Treatment for Water Reuse", Water Research , 84 (2015), pp. 315-322.. http://dx.doi.org/10.1016/j.watres.2015.07.034